# Fertility and Insemination Characteristics of Sperm Storage Tubules in Old Thai-Native Hens

**DOI:** 10.3390/ani14050694

**Published:** 2024-02-23

**Authors:** Theerapat Kheawkanha, Vibuntita Chankitisakul, Maruay Pimprasert, Wuttigrai Boonkum, Thevin Vongpralub

**Affiliations:** 1Department of Animal Science, Faculty of Agriculture, Khon Kaen University, Khon Kaen 40002, Thailand; k.theerapat@kkumail.com (T.K.); p.maruay@kkumail.com (M.P.); wuttbo@kku.ac.th (W.B.); vthevi@kku.ac.th (T.V.); 2Network Center for Animal Breeding and Omics Research, Faculty of Agricultural, Khon Kaen University, Khon Kaen 40002, Thailand

**Keywords:** semen concentration, insemination frequency, fertility rate, Thai indigenous chicken

## Abstract

**Simple Summary:**

In chicken, sperm is transported through the cloaca to the uterovaginal junction, which consists of sperm storage tubules (SSTs) which store sperm for long periods and retain its fertilizing capacity. Declining fertility in old hens is speculated to be due to a higher sperm release rate from the SSTs of old hens compared to that of younger hens. To enhance older breeds’ fertility efficiency, the optimal number of sperm-per-insemination dose (150–250 × 10^6^ spz/dose) together with the insemination frequency (once, twice, and thrice weekly) and the morphological appearance of SSTs after insemination were explored in this study. The present study’s findings showed that the sperm concentration dosages did not influence fertility. The highest fertility rate was obtained through a thrice-weekly insemination frequency. The SST sperm content-related categories helped explain these results. The SST categories containing high levels of sperm decreased significantly in the once-weekly insemination group from day four onward. Meanwhile, only the thrice-weekly insemination group maintained a high sperm count in the SSTs throughout the week. In summary, an insemination dose of 150 × 10^6^ sperm was sufficient for fertilization, but a thrice-weekly insemination frequency was most appropriate for sustaining the fertility rate in aged Thai native hens.

**Abstract:**

We aimed to evaluate the effects of sperm concentration (150–250 × 10^6^ spz/dose) and insemination frequency (once, twice, and thrice weekly) on fertility and sperm storage tubule (SST) characteristics. The SSTs were classified into five categories: namely, SSTs having an unscorable (SST1), empty (SST2), low (SST3), medium (SST4), and high (SST5) sperm count after insemination. The results showed that only insemination frequency affected the fertility rate (*p* < 0.05). The highest fertility was found in the thrice-weekly insemination group; however, this rate was not significantly different from that for the twice-weekly insemination group, except on day 7, while the once-weekly insemination group showed the lowest fertility rate (*p* < 0.05) from day four onward. On day 1, the SST characteristics showed no differences among the various insemination frequencies. On day 4, the SST2 and SST3 categories increased in the once-weekly insemination group (*p* < 0.05), while the SST4 and SST5 categories decreased compared to the twice- and thrice-weekly insemination groups (*p* < 0.05). On day 7, only the thrice-weekly insemination group maintained a level of SST5 category tubules like that measured on day 1 (*p* > 0.05). In summary, the insemination dose of 150 × 10^6^ sperm was enough for fertilization, and thrice-weekly insemination was the appropriate frequency in old Thai native hens for maintaining a high sperm density in the SSTs throughout the week.

## 1. Introduction

Traditionally, one breed of Thai native chickens, Pradu Hang Dum (PD; Figure 1), is raised in backyard farms in rural households, mainly for home consumption [1]. Distinct features of its meat, in terms of nutrients (higher protein and nutritional components, lower fat), textures, and flavor, make it superior to commercial broiler meat and now widely consumed [2,3]. Consequently, semicommercial-scale production of this breed is gradually increasing to meet the extensive demands for poultry. In commercial breeding flocks, most hens are used for 73–80 weeks, as their egg production decreases after this time [4,5]; meanwhile, Thai native hens are raised for longer periods of time [6]. Therefore, a study on extending the reproductive performance of old Thai native hens for longer effective utilization is interesting.

A decrease in flock fertility in old hens depends on many factors, including a decline in the sperm storage capacity of the oviduct [7]. Sperm storage within the female reproductive tract is necessary to maintain fertility, specifically asynchronous copulation and ovulation, in all animal species [8]. In chickens, sperm is transported through the cloaca to the uterovaginal junction (UVJ), which consists of sperm storage tubules (SSTs) which store sperm for long periods of time and retain its fertilizing capacity [9,10]. Previous studies have demonstrated that, during the breeding season, hens from the same flock with a low fertilization rate have less sperm retention within their fallopian tubes than those with a high fertilization rate [11,12]. Similarly, Van Krey and Leighton [13] showed that turkeys with normal fertility possessed sperm within the UVJ > 24 h post-insemination, while the UVJ of infertile hens did not have any sperm. The ability to retain sperm may depend on the characteristics of the SSTs.

The destruction of SSTs, including swelling in older hens brought about by long-term use for artificial insemination (AI), could be the reason for the decreased ability of SSTs to preserve sperm [14]; however, this phenomenon has been observed even in virgin old hens [6]. Therefore, the reason for declining fertility in old hens is speculated to be a higher sperm release rate from the SSTs of old hens compared to those of younger hens, as mentioned in unpublished data by Brillard [7]. This was confirmed by Kheawkanha et al. [6], who demonstrated that the percentage of sperm-containing SSTs 24 h after artificial insemination in mature hens was higher than that in old hens. Therefore, additional management practices may be required for old hen flocks to regulate their ability to be fertilized and increase their fertility after insemination.

An adequate number of sperm cells together with a greater frequency of internally stored sperm for longer periods should be considered. In the former strategy, Burrows and Quinn [15] reported that insemination with either 0.05 mL of diluted semen or 0.10 mL of undiluted semen resulted in a high fertility rate. Sperm doses of 50–200 million are required to produce the most fertile eggs in hens and turkeys [16,17,18]. However, less than 2% of an inseminated dose of 100 to 200 million sperm was found in the SSTs of either turkeys or chicken [19]. This suggests that a high sperm dose is not related to high fertility [20]. Meanwhile, previous research has mentioned that the factor affecting sperm content in SSTs is the insemination frequency and not the insemination dose [21]. The results of housing two cocks in a suitable mating ratio could increase mating frequency within the flock and increase the fertility of Thai native hens [22]. Thus, in old hens, inseminations performed frequently with a moderate number of sperm cells might be more efficient than inseminations performed with higher doses over longer intervals. However, such studies on artificial insemination and the morphological appearance of SSTs in old hens have not been evident. Therefore, in this study, we aimed to determine the fertilization capacity following artificial insemination with different sperm concentration dosages (150 and 250 × 10^6^ spz/dose) and insemination frequencies (once, twice, and thrice weekly) and characterize the morphological characteristics of the sperm storage reservoirs in terms of resident sperm in the UVJ in old Thai native hens. These results may provide a way to determine the optimal sperm concentration dosage and insemination frequency in old Thai native hens, thereby prolonging the lifespan of the breed stock and enhancing its efficiency, even in older breeds.

## 2. Materials and Methods

### 2.1. Animal and Management

All the animals in this study were procured from the Network Center for Animal Breeding and Omics Research (NCAB) (Khon Khan University, Khon Kaen, Thailand), which has standardized farm management and production systems according to the requirements of the Department of Livestock Development (Record No. 644001GFM0631).

One hundred and twenty Thai native hens (Pradu Hang Dum) at 73–75 weeks of age and with a weighted average of 2.01 ± 0.20 kg were selected for the experiments. In an open-environment housing system, they were placed in individual cages (48 × 45 × 45 cm). They were fed approximately 110 g of a commercial feed (Balance 924, Betagro Company Limited, Thailand) that provided 17% protein, 3% fat, 6% fiber, and 13% moisture. Water was provided ad libitum.

Twenty Thai native roosters (Pradu Hang Dum) at 40–52 weeks of age and a weighted average of 3.51 ± 0.23 kg were used for semen collection. The roosters were placed in an individual cage (60 × 45 × 45 cm). They were fed approximately 130 g of commercial feed, the same as the hens, and were provided with water ad libitum.

### 2.2. Experimental Design and Sample Collection

To determine the effect of an optimal sperm concentration dosage and insemination frequency on the fertility in old hens, the hens were randomly divided into six groups (n= 20 per group) incorporating various insemination factors, which included sperm concentrations (150 and 250 × 10^6^ spz/dose) and insemination frequencies (once on day 0; twice on day 0 and 3; and thrice on day 0, 3, and 6), as shown in Figure 2a. The eggs were collected during a 7-day period (starting from day 2 to day 8 after insemination) for incubation to determine the fertility (%). After egg collection for seven days, the hens rested for five days before starting another replication. The fertility test was replicated three times.

Upon completion of the fertility test, the hens rested for a week. The hens in each group were inseminated as in a previous study. They were slaughtered by cervical dislocation on days 1, 4, and 7, as shown in Figure 2b, to assess the resident sperm in the SSTs after insemination (n = 6 per group per time point). The percentage of sperm-containing SSTs was determined by tissue staining with hematoxylin and eosin.

### 2.3. Rooster Sperm Collection and Evaluation and Semen Dilution

Semen was collected using the dorsal abdominal massage technique [15]. The individual ejaculate was collected in a 1.5 mL microtube containing 0.1 mL of IGGKph diluent, as described by Chankitisakul et al. [23]. The sperm samples were transferred to the laboratory within 20 min of semen collection. The semen samples were examined under a microscope and assessed for the following criteria: mass motility, approximately 3 µL of semen sample was dropped onto the slide without a coverslip, examined immediately under a phase-contrast microscope at 40× magnification, and assessed on a scale of 1–5 (0 = no sperm movement; 5 = observable rapid waves and whirlpools, with more than 90% of sperm cells moving forward). The mass movement of the sperm required a score of ≥4 for inclusion in our experiment.

Sperm concentration was determined using a hemocytometer counting chamber (Lo-Laboroptik Ltd., Lancing, West Sussex, UK). Five microliters of semen sample was diluted with 995 µL of 4% sodium chloride solution. A drop of the diluted sperm sample was placed on the hemocytometer, and the sperm cells were counted under a compound microscope at 400× magnification. The semen samples that passed the evaluation criterion were required to have a concentration of ≥3 × 10^9^ sperm/mL.

For evaluating sperm viability and morphology, 20 µL of eosin-nigrosine stain solution was placed via a micropipette (Proline^®^ pipette, Sartorius, AG, Göttingen, Germany) on a defatted, sterilized microscope slide, and then a drop of 10 µL semen was added. These components were gently mixed for 30 s on a glass slide edge before preparing the smears. The glass slide was maintained at 37.5 °C until it dried completely. The eosin-nigrosine stain solution and microscope glass slides used in sample preparation were warmed to 37.5 °C on a heating plate. The stained slides were assessed using a phase-contrast microscope (Olympus, Tokyo, Japan) under an oil-immersion objective at 1000× magnification, and 200 sperm cells were counted. An unstained sperm cell was classified as “live sperm”. In contrast, a stained sperm cell with eosin was classified as “dead sperm”, and morphological normalcy was assessed by the typical spindle-shaped head and well-marked acrosome. The semen samples that passed the evaluation criteria were required to have a ≥85% sperm viability and a normal morphology.

Following semen examination, the semen samples that passed the evaluation requirements were pooled to increase semen volume and eliminate individual differences. The semen was then divided into two aliquots and diluted with IGGKph to the final concentrations of 150 × 10^6^ spz/dose and 250 × 10^6^ spz/dose for insemination.

### 2.4. Fertility Test

The artificial insemination method was performed as described by Chankitisakul et al. [23]. Briefly, fertilizing capacity was tested using intravaginal insemination of the hens at days 0 (once-weekly artificial insemination), days 0 and 3 (twice-weekly artificial insemination), and days 0, 3, and 6 (thrice-weekly artificial insemination) with a dose of 0.2 mL of semen (150 × 10^6^ spz/dose and 250 × 10^6^ spz/dose). Insemination was performed between 3:00 and 5:00 pm. Eggs collected from days 2 and 8 after insemination were incubated in an incubator, and fertility was determined by candling eggs on day 7 of incubation. Fertility was calculated as the percentage of fertile eggs relative to the total eggs.

### 2.5. Characteristics of Reproductive Organs and UVJ Tissue Preparation

The animals were slaughtered by manual cervical dislocation. This method consisted of stretching the chicken’s neck and then dislocating the joint between the skull and the spine at the first cervical vertebra by rotating the bird’s head upwards into the back of the neck. After the chicken stopped breathing, the entire reproductive system (ovary to vent) was carefully removed. The weights of the ovary and oviduct (emptied of contents) were measured using a weighing balance (Adam NBL214i, Adam Equipment, Oxford, CT, USA) and photographed to measure the oviductal length (cm) using image analysis software (ImageJ v. 1.8.0 software, NIH Image, Bethesda, MD, USA). The mean (±SEM) values for the body weight, ovary weight, oviductal weight, and oviductal length of the old hens were 2.13 ± 0.17 kg, 42.27 ± 16.54 g, 45.24 ± 11.56 g, and 74.64 ± 11.42 cm, respectively.

The UVJ tissues were prepared as previously described [14]. Briefly, the UVJ tissues were identified and dissected under a stereomicroscope, where they appeared as distinct bands of thin and convoluted folds [24]. The tissues were immediately fixed in 10% formalin in PBS for 24 h, embedded in paraffin, and cut with a thickness of 6 µm. Two longitudinal sections of UVJ were prepared for each hen. The sections were stained with hematoxylin and eosin on glass slides. The sections were examined under a light microscope using computer-assisted software for image analysis (Aperio ImageScope v. 12.4.6, Leica Biosystems, Deer Park, IL, USA).

### 2.6. Percentages of SSTs Containing Sperm

All SSTs presented on each slide were examined for the presentation of sperm. The SSTs containing sperm were divided into two groups (low and high sperm density), as described by Pierson et al. [25], with a few modifications, as shown in Figure 3. SST1 corresponded to an SST that could not be identified (unscorable; Figure 3a); SST2 was an SST without sperm (empty; Figure 3b); SST3 corresponded to an SST with 1–5 sperm cells (low; Figure 3c); SST4 was an SST containing 6–20 sperm cells (medium; Figure 3d); and SST5 was an SST with more than 20 sperm cells (high; Figure 3e). The data were expressed as the percentage of SSTs with sperm for each group.

### 2.7. Statistical Analysis

The Proc UNIVARIATE procedure using the SAS v. 9.0 software allowed us to examine the data distribution, which included assessing the normality using the Shapiro–Wilk test and determining that a normal distribution corresponded to *p* > 0.05 [26,27], conducting a homogeneity of variance test using the Levene’s test, and checking data outliers (±3SD was defined as an outlier), if any, that had not been included in our statistical analysis. After editing, the sperm concentration dosages and the insemination frequencies for the fertility test together with the insemination frequencies and the day of UVJ tissue collection for the SSTs containing sperm test were analyzed using a split-plot design (main plot RCBD) following a general linear model (GLM) procedure in SAS (SAS Institute, Inc., Cary, NC, USA). The observation values are expressed as mean values ± SEM. The factors and their interactions were compared using Tukey’s post hoc test. The overall differences between the treatment means were considered significant when *p* < 0.05.

## 3. Results

### 3.1. Fertilization Capacity

The levels of significance of the influence of sperm concentration dosages and insemination frequencies on the fertility rate are shown in Table 1. The interaction effect between the sperm concentration dosages and the insemination frequencies was not insignificant (*p* > 0.05). Only the insemination frequencies effect was significant on the fertility rate (*p* < 0.001); meanwhile, the different sperm concentration dosages did not influence the fertility (*p* > 0.05).

The results of the fertilization capacity after insemination with different sperm concentration dosages (150 and 250 × 10^6^ spz/dose) and insemination frequencies (once, twice, and thrice weekly) are shown in Table 1. The highest fertility rate was obtained in the thrice-weekly insemination group but did not differ significantly from the twice-weekly insemination group (*p* > 0.05), except on day 7 of egg collection. Meanwhile, the fertility rates were the lowest in the once-weekly insemination group (*p* < 0.05).

The hens’ daily egg production in the current study was 51.93 ± 15.58%. Figure 4 shows the daily fertility rates for the different insemination frequencies. The fertility rate tended to decrease continuously during the weekly inseminations. The fertility rates of the once-weekly insemination group were lower than those of the other groups from day 4 of egg collection onward (*p* < 0.05). The fertility rates of the twice-weekly and thrice-weekly insemination groups did not differ, except on day 7 of egg collection (*p* < 0.05).

### 3.2. Characterization of SSTs Containing Sperm after 24 h from Artificial Insemination

Based on the results of Table 1 and Figure 4, which show the influence of the insemination frequencies on the fertilization capacity, we assessed whether the insemination frequency groups differed in the resident sperm content of their SSTs after insemination. For this purpose, we determined the association between the frequency of insemination, the day of UVJ tissue collection, and the SST resident sperm content characterized 24 h after artificial insemination on days 1, 4, and 7 (Table 2). The interaction effect between the insemination frequency and the day of UVJ tissue collection was significant in the SST2, SST3, SST4, and SST5 (*p* < 0.05) categories. Meanwhile, the SST1 category was not affected by the insemination frequency and the day of UVJ tissue collection (*p* > 0.05).

The effect of the insemination frequency and the day of UVJ tissue collection on the SSTs’ resident sperm content is shown in Figure 5. There was no significant difference in the resident sperm content of unscorable SSTs (SST1) among the three insemination frequency groups at any time. The percentages of SST1 ranged from 20.22 to 28.89%. On day 1, there were no differences among the insemination frequency groups for all SST characteristics; the mean percentage distribution of SST1, SST2, SST3, SST4, and SST5 was approximately 21.67 ± 2.01, 21.99 ± 1.85, 14.60 ± 1.61, 21.12 ± 1.48, and 19.14 ± 1.63%, respectively.

A significant effect of insemination frequency on the resident sperm content of the SSTs in categories SST2-SST5 was observed from day 4 onward. On day 4, the SSTs without sperm (SST2) and the SSTs with 1–5 sperm cells (SST3) increased in the once-weekly insemination group (30.31 ± 1.94 and 24.72 ± 3.19%, respectively; *p* < 0.05), and the SSTs containing more than 20 sperm cells (SST5) decreased (2.64 ± 0.32%; *p* < 0.05), while the number of SSTs having 6–20 sperm cells (SST4) was similar to the twice- and thrice-weekly insemination groups (*p* > 0.05). However, the distribution of all SST categories remained unchanged in the twice- and thrice-weekly insemination groups (*p* > 0.05); moreover, those SST categories’ distribution percentages were similar to the day 1 values (*p* > 0.05).

On day 7, the SSTs without sperm (SST2) of the once-weekly insemination group were significantly increased (32.15 ± 1.83%; *p* < 0.05), while the SSTs with medium and high sperm counts (SST4 and SST5) were significantly decreased (6.38 ± 1.34% and 0.75 ± 0.22%; *p* < 0.05) compared to the others. In the twice-weekly insemination group, the number of SSTs without sperm (SST2) was higher than that of the thrice-weekly insemination group (25.59 ± 2.55% and 21.15 ± 1.94%, respectively) but had no statistical significance. The number of SSTs belonging to the SST category with 1–5 sperm cells (SST3) in the twice-weekly insemination group was higher compared to that of the thrice-weekly insemination group (21.71 ± 2.49% and 14.72 ± 2.06%, respectively; *p* < 0.05). However, the amount of SSTs belonging to the SST category with 6–20 sperm cells (SST4) in the twice- and thrice-weekly insemination groups was not different (15.61 ± 2.03% and 17.64 ± 0.56%, respectively). The number of SSTs belonging to the SST category with >20 sperm cells (SST5) was highest in the thrice-weekly insemination group (14.19 ± 3.20%; *p* < 0.05). Additionally, the percentage distribution of all SST categories on day 7 was similar to that on day 1 in the thrice-weekly insemination group (*p* > 0.05).

## 4. Discussion

Assessing fertilization by considering the optimal sperm concentration dosage and insemination frequency is critical for maintaining fertility rates in poultry production. The present study examined the relationship between sperm concentration dosage, insemination frequency, and fertility. We also characterized the morphological characteristics of the sperm storage reservoir in terms of resident sperm in the UVJ 24 h after insemination in old native Thai hens. The results of this study suggest that only the insemination frequency influences fertility capacity. The highest fertility rate was obtained through a thrice-weekly insemination frequency, a rate which did not differ from the rate registered for the twice-weekly insemination, except on day 7 of egg collection. From day 4 of egg collection onward, the fertility rates of the once-weekly insemination group were lower than those of the other groups. The SST sperm content-related categories were useful in explaining these results. The SSTs containing high levels of sperm decreased significantly in the once-weekly insemination group from day 4 onward. However, SSTs with a high sperm content could be maintained after day 4 if the insemination frequency was twice or thrice weekly.

Ensuring an adequate sperm accessibility for fertilization offers apparent advantages for the female. It has been discovered in domestic hens that sperm numbers, type of hen (broiler or layer), and age may all affect sperm storage in vivo and, consequently, egg fertility [21,28]. Sperm concentrations of 150–250 × 10^6^ are sufficient to maintain a satisfactory fertilization rate in chicken and turkey [29,30]. However, we discovered that sperm concentration (150 to 250 million/dose) did not affect the fertility rate; only insemination frequency (once, twice, and thrice weekly) affected fertility. Previous reports have also shown, similarly to our results, that there was no significant interaction between fertility and the fertile period when inseminating Isa Brown pullets with various sperm dosages (50, 100, and 150 million/0.1 mL) [31] or when using a sperm dose range of 15 to 60 million in Japanese quails [32]. Moreover, there was no significant increase in fertility with higher sperm dosages (5–9.6 million sperm cells) in ostriches [33]. This may result in a sperm selection mechanism in the female reproductive tract, specifically in the vagina, which is considered the main site for sperm selection [19,33]. Upon artificial insemination or natural mating, only 1% of chicken [7] and 2% of turkey [34] sperm of a high quality enters the SST within 24 h of insemination. The remaining sperm is either ejected from the oviduct or degraded [35]. This may indicate that good fertility depends not on the concentration of semen used for insemination but on whether the number of sperm cells retained in the SST is sufficient for fertilization.

Once sperm cells are in the storage site, the appropriate functioning of SSTs is thought to be critical for fertility, and a loss in sperm storage capacity has been correlated with fertility senescence in birds [36]. The significant differences in the fertile periods observed between species have been related to changes in SST-specific sperm numbers and sperm storage capacity [37]. According to Bakst et al. [38], the biological basis of sustained fertility in chicken and turkeys is the ability of sperm to reside in SSTs. Differences in fertility duration in their study depended on the species, with that of domestic fowls (2–3 weeks) and turkeys (10–15 weeks) being related to their respective number of SSTs (4893 and 30,566 SSTs in chicken and turkeys, respectively). The status of the reproductive tract in hens is important for sperm storage. The fertility period decreases as the rate of sperm release from storage increases [39]. However, it is evident that older hens exhibit a more rapid release of sperm [40] and sperm loss in their SSTs than younger hens [41]. The exact mechanism of sperm release is unclear, and how sperm cells enter and exit the SSTs remains poorly understood; however, it has been reported recently that not all but just a portion of the sperm is released from the SSTs to the infundibulum during ovulation because SSTs have a constricted “gate-like” entrance and act as a physical barrier, preventing sperm from leaving [42]. It has been speculated that the gate-like narrowed openings for sperm selection in older hens are not fully functional. Our previous study demonstrated the destruction of the SSTs found in virgin old Thai native hens [6], and long-term use resulted in swelling of the SSTs in Rhode Island Red hens [14], a phenomenon which may be associated with the conformational changes in the SSTs’ shape, from functional to nonfunctional, explaining the decreased sperm reservoir within the SSTs in old hens. Thus, the number of sperm cells remaining stored in the SSTs of old hens for further insemination decreases. This might explain why older hens with shorter fertile periods had the most significant percentage of SST sections containing sperm only 2 days after insemination in [21] and why there was a significantly lower infertility rate after 2 days of egg collection in [6].

Therefore, it is necessary to inseminate hens more frequently to fill their SSTs continuously. Our experimental findings confirmed that once-weekly insemination could not maintain SSTs containing a high sperm density for a week in old hens. The number of SSTs without sperm (SST2) and with low sperm counts (SST3) significantly increased on day 4. Even the number of SSTs containing a medium sperm density (SST4) did not differ from the other insemination frequency groups; however, the number of SSTs with a high sperm density (SST5) was significantly lower in the once-weekly insemination group, suggesting that a high sperm content in the SSTs is necessary for fertility. A similar situation occurred with twice-weekly insemination on day 7 of egg collection; the number of SSTs containing a medium sperm density (SST4) did not differ from that of the thrice-weekly insemination group; however, the percentage of fertility on day 7 of egg collection was different, and this resulted in a lower fertility rate on day 7 of egg collection in the twice-weekly insemination group compared to the thrice-weekly insemination group. SSTs containing more than 20 sperm cells are necessary for higher fertilization rates. It might be suggested that more frequent inseminations together with a high sperm content in SSTs are necessary for fertility potential. This is supported by Brady et al. [43], who examined molecular level changes in the SSTs of turkeys and demonstrated that the function of the SSTs is impacted by the presence of sperm in the SSTs and the act of the insemination itself, correlating in such a wat as to promote fertility ability. Additional research is needed to clarify the exact cellular processes behind these factors that take place in SSTs during insemination in chicken.

## 5. Conclusions

The results of this study suggest that an insemination dose of 150 × 10^6^ sperm was sufficient for fertilization, while the most appropriate insemination frequency for maintaining an optimal fertility rate and sustaining SSTs with a high sperm density throughout the week in old Thai native hens was three times weekly.

## Figures and Tables

**Figure 1 animals-14-00694-f001:**
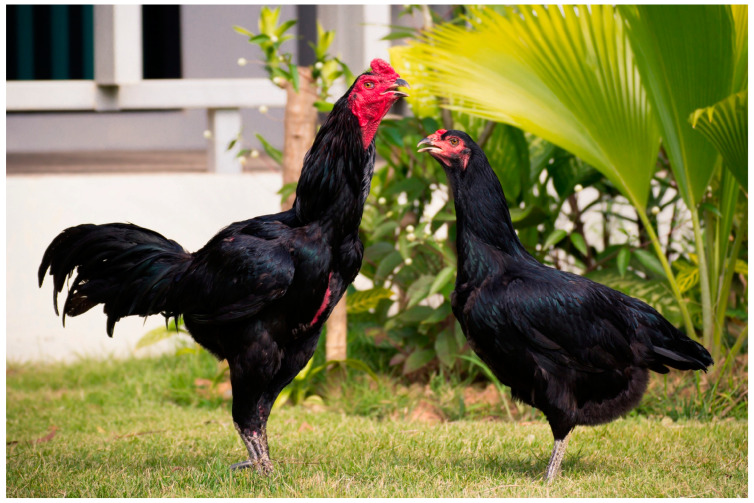
Morphology of Pradu Hang Dum chickens: male (left side) and female (right side). Overall, the ideal traits are strong, with the feathers covering most of the body being black, with red-to-dark brown feathers on top, mostly black eyes, black ear lobes, yellow and black beaks, red face and pea comb, and black shanks. The dark brown-to-black toes and claws should ideally be black.

**Figure 2 animals-14-00694-f002:**
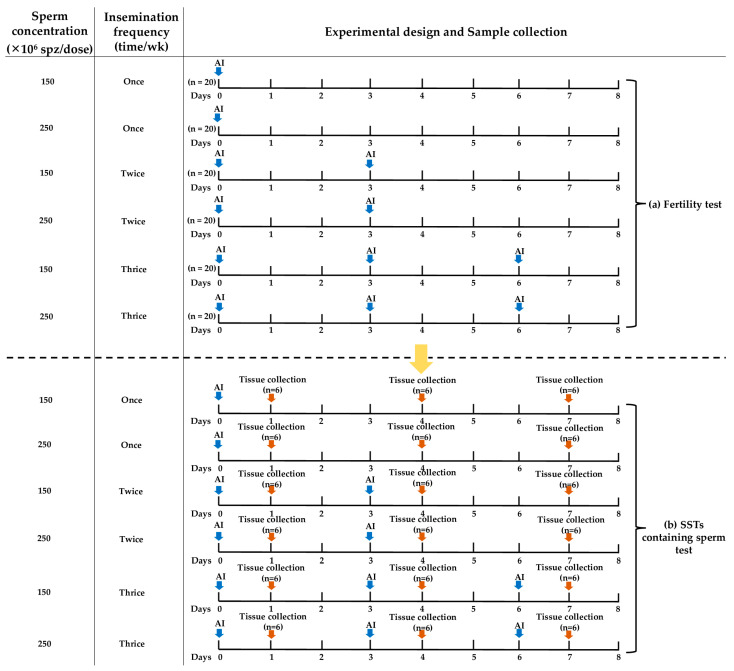
Experimental design and sample collection. The optimal sperm concentration dosage and insemination frequencies were examined by a fertility test (**a**). After finishing the fertility test, the resident sperm in the UVJ after insemination was examined by means of the sperm-containing SST test (**b**).

**Figure 3 animals-14-00694-f003:**
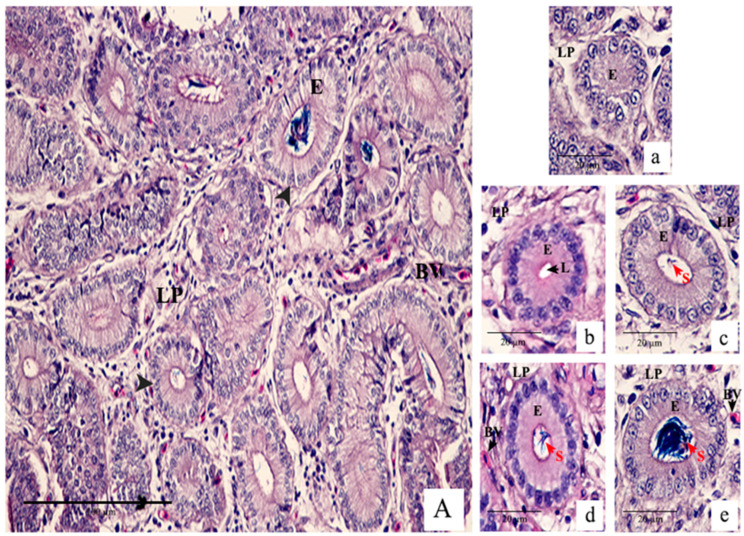
Transverse section of the uterovaginal junction (UVJ) fold of an old Thai native hen after insemination for at least 24 h containing sperm storage tubules (SSTs; black arrowhead) stained with hematoxylin and eosin (H & E) (**A**). Classification of SSTs within the UVJ section for sperm storage was assessed under a digital pathology scanner with computer-assisted software for image analysis. The samples were categorized into five groups: SST1 consisting of SSTs that could not be identified (unscorable; (**a**)), SST2 without sperm (empty; (**b**)), SST3 with 1–5 sperm cells (low; (**c**)), SST4 containing 6–20 sperm cells (medium; (**d**)), and SST5 with more than 20 sperm cells (high; (**e**)), respectively. S, sperm. Scale bars = 100 µm (**A**) and 20 µm (**a**–**e**). BV—blood vessel; E—epithelium; L—lumen, LP—lamina propria; and S—stoma.

**Figure 4 animals-14-00694-f004:**
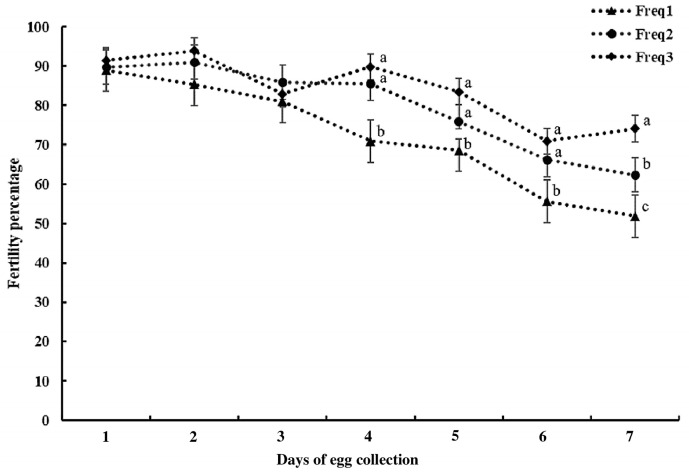
Fertility percentage (%) ± SEM comparison among different insemination frequencies (once, twice, and thrice weekly). ^a,b,c^ Different letters within a day after artificial insemination indicate significant differences (*p* < 0.05). The values are percentages of fertilized eggs per group and collection day. Freq1 = once-weekly insemination; Freq2 = twice-weekly insemination; and Freq3 = thrice -weekly insemination.

**Figure 5 animals-14-00694-f005:**
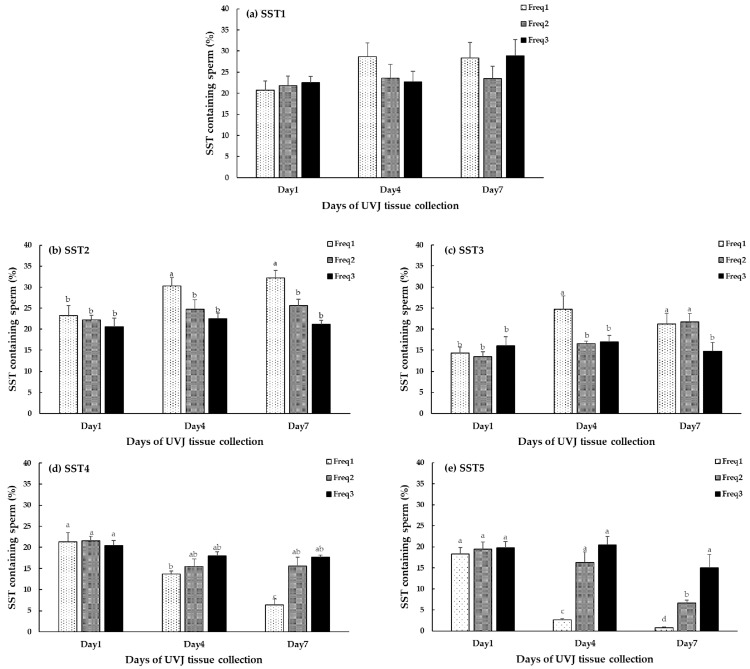
Effect of insemination frequency on SST sperm content characteristics in UVJ tissue after insemination at day 1, day 4, and day 7. ^a,b,c,d^ different letters indicate significant differences within each SST category (*p* < 0.05). (**a**) SST1 = SSTs containing sperm that could not be identified; (**b**) SST2 = SSTs without sperm; (**c**) SST3; SSTs with 1–5 sperm cells; (**d**) SST4; SSTs with 6–20 sperm cells; (**e**) SST5; SSTs with more than 20 sperm cells. Freq1; once-weekly insemination; Freq2; twice-weekly insemination; and Freq3; thrice-weekly insemination.

**Table 1 animals-14-00694-t001:** Effect of different sperm concentration dosages (150 and 250 × 10^6^ spz/dose) and insemination frequencies (once, twice, and thrice weekly) on the fertility rate in old Thai native hens.

Parameters	Groups of Artificial Insemination						SEM	Effect (*p* Values)		
	150 × 10^6^ spz/dose			250 × 10^6^ spz/dose						
	Once	Twice	Thrice	Once	Twice	Thrice		Conc.	Freq.	Conc. × Freq.
	Weekly	Weekly	Weekly	Weekly	Weekly	Weekly				
Number of eggs	224	221	221	205	218	216	-	-	-	-
Total fertility (%)	65.78 ^b^	78.07 ^a^	83.68 ^a^	76.59 ^b^	78.87 ^a^	84.06 ^a^	7.49	0.2727	0.0419	0.0551

^a,b^ Different letters within a row indicate significant differences (*p* < 0.05). SEM = standard error of the means; Conc. = sperm concentration dosages; Freq. = insemination frequencies; and Conc. × Freq. = interaction between sperm concentration dosages with insemination frequencies.

**Table 2 animals-14-00694-t002:** Analysis of variance summary for different characteristics of SSTs containing sperm in UVJ after insemination with different frequencies of insemination and day of UVJ tissue collection on days 1, 4, and 7.

Source of Variations	Characteristics of SSTs Containing Sperm (*p* Values)				
	SST1	SST2	SST3	SST4	SST5
Freq.	0.7095	0.6064	0.0312	0.0003	0.0001
Day of UVJ tissue collection	0.3155	0.4443	0.0025	0.2780	0.9043
Freq. × Day of UVJ tissue collection	0.4413	0.0424	0.0326	0.0002	0.0001

Freq. = insemination frequencies; SST = sperm storage tubules; SST1 = SSTs containing sperm that could not be identified; SST2 = SSTs without sperm; SST3 = SSTs with 1–5 sperm cells; SST4 = SSTs with 6–20 sperm cells; and SST5 = SSTs with more than 20 sperm cells.

## Data Availability

Data are contained within the article.

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
