# Peer review of "Fertility and Insemination Characteristics of Sperm Storage Tubules in Old Thai-Native Hens"

_animals, 2024, doi:10.3390/ani14050694_

Round 1

Reviewer 1 Report

Comments and Suggestions for Authors

The research, titled "Fertility and Insemination Characteristics of Sperm Storage Tubules in Old Thai Native Hens," aims to evaluate the effects of different sperm concentrations and insemination frequencies on reproductive performance in Thai chickens. The manuscript aligns with the scope of the journal, and given the scarcity of data on poultry breeds, it may be of interest to readers.

However, there are several areas that could benefit from improvement:

The simple summary needs to be rewritten to adhere to the journal's instructions for authors (https://www.mdpi.com/journal/animals/instructions). It should include a clear statement of the problem addressed, the aims and objectives, pertinent results, conclusions from the study, and their societal value. Additionally, it should be written for a lay audience.

I recommend rewriting the abstract to enhance readability. Consider including more numerical data with corresponding significance levels to strengthen the abstract's content.

Given the unique characteristics of the breed, it is advisable to include more details on the morphological and rearing characteristics of these chickens in the introduction section.

Provide more details on the subjects used in the research. For instance, what was the average weight of the hens? Where did the animals come from? Were they born in the same incubator and from the same group of breeders? Include information on the characteristics of the diet administered to the animals.

To enhance the transparency and replicability of your research, I kindly suggest that you include a section detailing the methods employed for dietary analysis. This should encompass the techniques and procedures used to determine the composition of the diet. Furthermore, I recommend referencing a reputable source for these methods, such as the protocol outlined in 10.3390/vetsci10090554.

The sample size and experimental design should be discussed in more detail. Providing information on the number of animals used, their characteristics, and the statistical methods employed would enhance the study's transparency and replicability.

The statistical methods described in the manuscript may benefit from further elaboration and clarity. It would be helpful to provide more details on the specific statistical tests employed, their rationale, and how they align with the research questions.

Could you please clarify whether you conducted tests for normality and homogeneity on your data before proceeding with the statistical analysis? It's crucial to ensure that the assumptions underlying your chosen statistical methods are met. I recommend referring to the guidelines outlined in [proposed reference, e.g., 10.1186/s12917-022-03289-2 and 10.3390/ani12141740] for conducting such tests to maintain the rigor and reliability of your analysis.

Explain how the data were presented and whether any transformations or adjustments were made to the raw data. Clarify how outliers, if any, were handled in the analysis.

To facilitate transparency and future research, consider sharing the data and detailed methodology used in this study.

Enrich the discussion by addressing the study's limitations and practical applications.

In addition to the current findings, a potential avenue for future improvement in understanding the reproductive processes in old Thai native hens could involve the study of sperm lipidomics. Sperm lipidomics investigates the composition and changes in lipid profiles within sperm cells, which could provide valuable insights into fertility-related mechanisms.

For example, lipidomics analysis could focus on identifying specific lipid molecules, such as phospholipids, glycerolipids, and sphingolipids, in sperm cells. Changes in the abundance or composition of these lipids could be correlated with fertility outcomes and SST characteristics. Lipidomics studies may also shed light on the role of lipids in membrane integrity, cellular signaling, and other processes crucial for sperm function and storage. (see: 10.3390/ANI13010008)

Furthermore, exploring the impact of dietary or environmental factors on sperm lipid composition in relation to insemination frequency and sperm concentration may offer a more comprehensive understanding of the factors influencing fertility in old Thai native hens. Integrating lipidomics data with the current findings could enhance the overall understanding of the reproductive biology of these hens and contribute to the development of targeted strategies for improving fertility outcomes.

In summary, incorporating sperm lipidomics into future research endeavors could deepen our understanding of the molecular aspects underlying the observed effects of insemination frequency and sperm concentration on fertility and SST characteristics in old Thai native hens.

I recommend incorporating a discussion paragraph highlighting the significance of educating future veterinarians, technicians, and farmers about the issues addressed in the paper. Emphasizing the importance of effective teaching methods in shaping knowledgeable students and proficient equine veterinarians would add depth to the paper's implications. It is advisable to refer to recent publications on veterinary education to provide up-to-date insights into best practices in preparing future professionals to address the challenges discussed in the paper. Please see: 10.1016/j.jevs.2023.104537 and 10.3390/ani13223503.

Please double-check the reference list to ensure that all references are included in the main text and vice versa.

Reviewer 2 Report

Comments and Suggestions for Authors

 The article is well written; the title and the abstract reflect the contents of the paper, and the introduction clearly states the problem being investigated and provides an adequate background. The methodology is adequate, the results are well presented and the discussion is sound. 

1Introduction: the background literature is rather old.

2. The manuscript requires English polishing.

3. Please provide a detailed description of the chicken's execution method. In addition, indicate whether the method meets ethical standards.

Comments on the Quality of English Language

The English writing needs to be revised to make it more readable.

Reviewer 3 Report

Comments and Suggestions for Authors

The MS entitled Fertility and insemination characteristics of sperm storage tubules in old Thai native hens” by Kheawkanha et al investigated the effects of sperm concentration and insemination frequency on fertility and sperm storage tubules (SSTs) characteristics. The authors found that an insemination dose of 150 x 106 sperm was sufficient for fertilization and a thrice weekly insemination frequency was most appropriate to sustain the fertility rate in aged Thai native hens. Overall, this study is of great significance for the utilization of roosters in poultry production.

General comments

Although the authors have applied a figure to demonstrate the procedure of this study, the methods are not well corresponding to the text. For example, whats the time of sampling? After continuously 3 weeks insemination? Or immediately after each insemination?

Regarding to the Percentages of SSTs containing sperm, how many sections were evaluated? Are the sections continuously or selected? what kind of sperm storage tubules were selected? All these should be demonstrated in the method part.

Regarding the statics method and description: for the total fertility (%), maybe the X2 is appropriate for the statistics; for the fertility percentage of multiple comparison, different letters should be used in Figure 3.

The English writing needs to be moderately improved.

Specific comment

L13: compared to:  use  withinstead of to

L15:  ”appropriate is redundant

L42: that consists of :  use  whichinstead of that

L99 continuously for 3 weeks. Should be included in the figure 1

L103-104: on days 1, 4, and 7. It means three weeks later or continuously 3 weeks later?

L185 figure 2 fig legend: add the notes for the abbreviations in the picture

L214: were not influenced by fertility: did not influence the fertility

L221: when was the fertility tested? after three weeks insemination? Corresponding to the methods part and Figure 1.THE SAME AS FIGURE 3

Comments on the Quality of English Language

 Extensive editing of English language required.

Round 2

Reviewer 1 Report

Comments and Suggestions for Authors

Good job.

Reviewer 3 Report

Comments and Suggestions for Authors

The authors have addressed the issues and improved the MS appropriately.